# Cancer-Associated Dysregulation of Sumo Regulators: Proteases and Ligases

**DOI:** 10.3390/ijms23148012

**Published:** 2022-07-20

**Authors:** Nieves Lara-Ureña, Vahid Jafari, Mario García-Domínguez

**Affiliations:** Andalusian Centre for Molecular Biology and Regenerative Medicine (CABIMER), CSIC-Universidad de Sevilla-Universidad Pablo de Olavide, Av. Américo Vespucio 24, 41092 Seville, Spain; nieves.lara@cabimer.es (N.L.-U.); vahid.jafari@cabimer.es (V.J.)

**Keywords:** SUMO, protease, ligase, cancer, regulation, transcription

## Abstract

SUMOylation is a post-translational modification that has emerged in recent decades as a mechanism involved in controlling diverse physiological processes and that is essential in vertebrates. The SUMO pathway is regulated by several enzymes, proteases and ligases being the main actors involved in the control of sumoylation of specific targets. Dysregulation of the expression, localization and function of these enzymes produces physiological changes that can lead to the appearance of different types of cancer, depending on the enzymes and target proteins involved. Among the most studied proteases and ligases, those of the SENP and PIAS families stand out, respectively. While the proteases involved in this pathway have specific SUMO activity, the ligases may have additional functions unrelated to sumoylation, which makes it more difficult to study their SUMO-associated role in cancer process. In this review we update the knowledge and advances in relation to the impact of dysregulation of SUMO proteases and ligases in cancer initiation and progression.

## 1. Introduction

Post-translational modifications (PTMs) are key regulators of most biological processes. Besides phosphorylation, methylation, acetylation and others, covalent modification of proteins by small polypeptides of the ubiquitin-like modifiers (UBLs) family have gained importance in recent decades. Among UBLs, the small ubiquitin-like modifier (SUMO), of ~90 amino acids and discovered in the nineties, has proven to regulate most cellular processes [1].

Up to five SUMO paralogs have been described in vertebrates. While SUMO4–5 present restricted patterns of expression and it is not clear whether they can be functionally conjugated to proteins [2], SUMO1-3 are widely expressed in all tissues and involved in modification of thousands of proteins [3,4,5]. SUMO2 and SUMO3 are virtually indistinguishable and normally designated as SUMO2/3. They are abundant in the cell in the unconjugated form and rapidly attached to proteins in response to a variety of stress stimuli [6,7,8]. By contrast, most SUMO1 is conjugated to proteins, and mainly to the nuclear pore protein RanGAP1 [9,10]. SUMO1 shares about 17% and 50% identity with ubiquitin and SUMO2/3, respectively. SUMO2 and SUMO3 are 97% identical. Sumoylation occurs at the ε-amino group of a Lys (K) residue, often included in the consensus I/L/VKxE/D. SUMO2/3 displays this consensus sequence, which facilitates the formation of poly-SUMO chains.

Despite being able to covalently modify other proteins, SUMO is also able to non-covalently interact with many proteins through SUMO interacting motifs (SIMs) present in interactors [11,12]. SIMs are of special relevance for sumoylation-dependent ubiquitination, through the action of SUMO-targeted ubiquitin ligases (STUbLs) like RNF4, which present tandem SIMs able to recognize poly-SUMO chains in ubiquitin targets to be degraded [13]. The combination of sumoylation sites with SIMs in the same or different proteins contributes to the formation of protein macrostructures which may enhance sumoylation by recruiting additional SUMO targets to a sumoylation favorable environment. For instance, the promyelocytic leukemia protein (PML) is deeply modified by SUMO, which in turn interacts with SIMs in PML giving rise to big aggregates (PML nuclear bodies (NBs)) recruiting additional sumoylated proteins and serving as a sumoylation platform for many other SUMO targets [14]. In this context, SUMO has been considered a molecular glue, and this is related to the concept of “protein group sumoylation”. This refers to the fact that sumoylation, in contrast to many other PTMs, frequently affects collectively to different proteins in a group of interacting proteins, rather than individually to a specific protein [15].

An enigmatic and premature observation on sumoylation is related to the fact that at any given time, only a very reduced fraction of the pool of a target protein appears modified by SUMO, while evidence suggests that the whole target pool is modified. For instance, the use of sumoylation mutants has frequently dramatic consequences, which is not expected if most of the molecules of a given protein make their function in the unmodified form and only a small percentage of molecules is modified for additional purposes. This paradox has been explained on the basis of sumoylation being, in most of the cases, a quite transient modification, but with permanent effects on protein function or destiny once SUMO has been removed [16].

Sumoylation is essential in vertebrates, and different knock out (KO) animal models support this. The unique conjugating enzyme of the sumoylation pathway (UBC9, see next section) has been demonstrated to be indispensable for survival of the mouse embryo. *Ubc9* KO mice embryos dye at the early post-implantation stage due to the inability of the blastocyst inner cell mass to expand, which enters in apoptosis [17]. Regarding SUMO paralogs, it has been described that both SUMO1 and SUMO3 are dispensable, probably due to compensation by the other SUMO molecules (reviewed in [18]). However, loss of SUMO2 cannot be compensated by any paralog, which seems to be related to the high abundance of SUMO2 in comparison with the other SUMO molecules [19].

SUMO attachment to proteins has a great impact on their functions, as it may alter localization, activity, stability, interactions and conformation of target proteins. SUMO is involved in regulation of most relevant cellular processes, and in particular in gene expression [20]. Thus, unbalanced sumoylation may lead to altered protein function or gene expression, resulting in cell transformation and tumorigenesis. Key regulators of the sumoylation process are the SUMO ligases and proteases, which determine the sumoylation status of target proteins. In this review, we summarize the current knowledge about dysregulation of these enzymes in cancer and their impacts on tumor initiation and progression.

## 2. The Sumoylation Pathway

The sumoylation pathway is quite similar to the ubiquitination pathway, but there is its own set of enzymes for modification by SUMO (Figure 1). This involves several steps: (i) initial maturation of the SUMO precursor, by proteolysis of several C-terminal amino acids to expose a Gly-Gly (GG) motif; (ii) activation of mature SUMO by the heterodimeric SAE1/UBA2 E1 enzyme, through ATP hydrolysis; and (iii) transfer to UBC9, the unique conjugating E2 enzyme in the sumoylation system. Both activation and transfer to UBC9 involves the formation of a thioester bond between the C-terminus of mature SUMO and a Cys (C) residue in the catalytic sites of E1 or E2, (iv) transfer from UBC9 to targets, frequently assisted by a SUMO ligase or E3. Besides, specific SUMO proteases are in charge of SUMO maturation and scission from targets.

### 2.1. SUMO Proteases

Proteases involved in SUMO maturation and recycling are SUMO specific, and the most studied are those of the SENP family (Figure 1 and Figure 2 and Table 1). This family includes SENP1–3 and SENP5–7 [21]. Besides, additional proteases also with specific activity on SUMO have been described more recently. They include a new family of desumoylating isopeptidases (DeSI) [22], which comprises DESI1 and DESI2, and USPL1 [23] and HINT1 [24] (Table 1). All SENP proteins display a desumoylation catalytic domain at the C-terminus, which appears split in the case of SENP6 and SENP7 (Figure 2A). Different sequences in the N-terminal region of SENPs have been related to cellular localization, and, in the case of SENP7, to interaction with the heterochromatin protein 1 (HP1) through two tandem PxVxL motifs [25]. Moreover, a SENP7 splice variant, lacking one of these motifs, has been reported to predict for good prognosis in breast cancer patients [26]. Additional variants have been described for other SENPs (https://www.ncbi.nlm.nih.gov/gene/?term=SENP (accessed on 14 July 2022)). SENP1 and SENP2 have been indicated to be able to maturate SUMO1–3, although SENP1 shows preference for SUMO1 and SENP2 for SUMO2 [21] (Figure 2B). SENP5 has been also shown to efficiently maturate SUMO2. Regarding SUMO recycling, SENP1 and SENP2 have been demonstrated to efficiently detach the three SUMO paralogs from targets, while SENP3 and SENP5 are more selective on SUMO2/3 and SENP6 and SENP7 display poly SUMO2/3 chains editing activity [21] (Figure 2B). In the cell, SENP1 and SENP2 have been associated with PML NBs in interphase (as SENP6) and with the nuclear pore complex (NPC); SENP3 and SENP5 with both the nucleolus and the mitochondria; and SENP6, SENP7 and SENP3 with chromatin [27] (Figure 2C).

### 2.2. SUMO Ligases

A key difference between SUMO proteases and ligases is that most SUMO ligases display additional functions independent of the SUMO ligase activity. This complicates in many cases the functional analysis in relation to the role of sumoylation. PIAS proteins (Figure 3), probably the most studied SUMO ligases, were among the first identified proteins displaying SUMO ligase activity, together with the NPC-associated protein RanBP2 and the Polycomb Repressive Complex 2 protein CBX4 (or PC2) [28]. Among these, PIASs and RanBP2 have been well characterized at the biochemical level (reviewed in [29]). The PIAS family comprises members 1–4. Two variants due to alternative splice have been classically described for PIAS2 (Figure 3A), although additional variants have been described for the different PIAS coding genes (https://www.ncbi.nlm.nih.gov/gene/?term=PIAS (accessed on 14 July 2022)). Structurally, PIAS proteins present an N-terminal SAP domain, involved in DNA binding as well as in interaction with transcriptional co-regulators, a PINIT region involved in nuclear localization, a RING-like SP-RING domain, an acidic stretch, and a C-terminal region rich in Ser/Thr (S/T) [30]. Different SIMs, relevant for PIAS function, have been described for the different members, both inside and outside the acidic stretch (reviewed in [31]). Most ligases are not required for *in vitro* modification of targets, which seems to depend exclusively on the presence of mature SUMO, E1 and E2. However, ligase requirement has been reported *in vivo* for a great variety of physiological processes. It was initially indicated that ligase activity in many cases relies in the ability of ligases to recruit at the same time, through different domains, SUMO-loaded UBC9 and the SUMO target to facilitate SUMO transfer, being the SP-RING domain the responsible for UBC9 recruitment in the case of PIAS proteins (Figure 3B). However, for RanBP2 ligase activity, target interaction seems to be dispensable, as minimal ligase appears to only require the simultaneous binding of SUMO1 and UBC9 to RanBP2 to optimally position the thioester bond for efficient transfer to its main target, RanGAP1 [32,33] (Figure 3B). Initial *in vitro* studies using the yeast PIAS1 ortholog Siz1 mapped the minimal E3 ligase domain to the region comprising the PINIT and the SP-RING motifs [34] (Figure 3B). Interestingly, another RING-related motif, the PHD domain, has been also implicated in UBC9 binding and sumoylation; for instance, in KAP1 and AtSIZ1 [35,36]. First studies on PIAS proteins also showed a significant level of promiscuity in target selection, although subsequent *in vivo* approaches have demonstrated more specific effects (reviewed in [18]). In contrast to ubiquitination that requires hundreds of E3 ligases for specific target selection, only a few dozens of ligases have been described for SUMO. Besides PIAS, RanBP2 and CBX4, enhanced sumoylation of specific targets has been associated with additional proteins, including: TOPORS, RSUME, MUL1, RHES, some proteins of the tripartite motif family (TRIM), ARF, SF2, class IIa histone deacetylases (HDACs), the PIAS-related proteins NSMCE2 and ZMIZ1-2, SLX4, KROX20, RNF212, UHRF2, TRAF7, ZBED1, MDM2 and ZNF451 (Table 1). Among TRIM proteins, SUMO ligase activity has been attributed to TRIM1, 11, 19, 22, 27, 28, 32, 33, 36, 38, 39 and L2 (Table 1). Interestingly, in some cases, as for TRIM27, and in relation to TP53, ligase activity has been reported for both SUMO and ubiquitin [37]. Additional proteins also show a dual function as E3 ubiquitin and SUMO ligases, such as UHRF2, TOPORS, TRAF7 or MDM2, and in some cases, distinct domains contribute to one or another activity [31]. Nevertheless, for some proteins dual function is controversial. For instance, TRIM25 has been described to stimulate TP53 sumoylation, but mechanistically this has been explained on the basis of TRIM25 requirement for recruitment of the RanBP2/G3BP2 complex, which ultimately mediates TP53 sumoylation [38]. Of note, ZNF451 has also been well characterized biochemically [39], and it has been proposed to display elongase E4 activity besides E3 ligase activity, due to its ability to assemble poly-SUMO2/3 chains through the action of tandem SIMs at the N-terminus [40]. Another open reading frame (ORF): KIAA 1586, close to ZNF451, encodes a protein with similar N-terminal tandem SIMs, which has also proved to display elongase activity. Known SUMO ligases are quite divergent phylogenetically and structurally and, as indicated, there are not defined families of proteins exclusively devoted to this function, which seems to appear in a high variety of proteins with additional functions. Thus, it seems likely that well-known proteins will reveal unexpected SUMO ligase activity in the near future.

**Table 1 ijms-23-08012-t001:** SUMO proteases and ligases.

Enzymes	Family	Proteins	References
**Proteases**	SENP	SENP1, SENP2, SENP3, SENP5, SENP6, SENP7	[21]
	USPL1	USPL1	[23]
	DeSI	DESI1, DESI2	[22]
**Ligases**	SP-RING	PIAS1, PIAS2, PIAS3, PIAS4	[28,30,31]
		NSMCE2 (NSE2/MMS21)	
		ZMIZ1 (ZIMP10), ZMIZ2	
	TRIM	TRIM1, TRIM11, TRIM19 (PML), TRIM22, TRIM27, TRIM28 (KAP1), TRIM32, TRIM33 (TIF1g), TRIM36, TRIM38, TRIM39, TRIML2	[37,41,42,43,44,45]
	HDACs IIa	HDAC4, HDAC5, HDAC7, HDAC9 (MITR)	[46]
	Elongases	ZNF451	[39,40]
		KIAA 1586	
	Others	RanBP2	[18,28,31,47,48,49]
		CBX4 (PC2)	
		TOPORS	
		RSUME	
		MUL1 (MAPL)	
		RHES	
		ARF (P14)	
		SF2 (ASF)	
		SLX4	
		KROX20	
		RNF212	
		UHRF2 (RNF107)	
		TRAF7 (RNF119)	
		BCA2 (RNF115)	
		ZBED1 (DREF)	
		MDM2	

Pathway components can be influenced by the physiological environment, which may have an impact in their activities. In this sense, several SENPs have been shown to be sensible to hypoxic inactivation [50]. In addition, SENP3 has been defined as a redox sensor, and, for instance, reactive oxygen species (ROS) may modulate E1 activity (reviewed in [51]). Besides, a number of PTMs, including phosphorylation, acetylation, ubiquitination and sumoylation itself, have been shown to affect several pathway components. Otherwise, a patent crosstalk between sumoylation and other PTMs like ubiquitination, phosphorylation and methylation operates on target modification [4,28,52]. As indicated, vertebrate viability depends on sumoylation, and acting on E1 or E2, which will globally affect sumoylation, should certainly have general and strong effects. However, the function of regulating modification of specific targets mostly relies on SUMO proteases and ligases. Target modification output depends on localization, expression, activity and paralog preference of these enzymes. Thus, their unbalanced expression, localization or function, directly impact on sumoylation of defined proteins, which in many cases is in the basis of physiological alterations leading to tumorigenesis. Depending on the specific association of particular ligases and proteases with precise targets, and, thereby, on the nature of these and the affected tissue, alterations will lead to the appearance of a great variety of cancer types.

## 3. SUMO in Cancer

Since SUMO participates in regulating many key cellular processes, it is understandable that unbalanced sumoylation is related to different pathologies. Important processes, the dysregulation of which is associated with genome instability, depend on SUMO. In fact, besides its prominent role in facing stress conditions, SUMO is tightly linked to cell cycle progression [53,54], which particularly associates SUMO dysregulation with cancer. The first link between SUMO and cancer is probably related to the oncogenic fusion of PML protein with the retinoic acid receptor alpha (RARα), which causes acute promyelocytic leukemia (APL), and actively involves SUMO in cell transformation [55].

Among the hallmarks of cancer are mutability, uncontrolled proliferation, resistance to growth suppression, immune evasion, inhibition of differentiation and apoptosis, angiogenesis, metastasis and metabolic alterations. A variety of processes related to these features depend on SUMO regulation. They include the DNA damage response (DDR), DNA replication, transcription, response to hypoxia, epithelial to mesenchymal transition (EMT), cell migration and invasion, apoptosis, immune response, inflammation, cell cycle regulation and cell division, telomere maintenance, stem cell-like properties maintenance and senescence [56,57,58]. Key pathways related to cancer are subjected to SUMO control, such as the cell cycle-associated PI3K/AKT/mTOR pathway, the immune response-associated NF-κB and JAK-STAT pathways, the mitogen-activated protein kinases/extracellular signal-regulated kinases MAPK/ERK cascade, TGFβ signaling and EMT pathway in metastasis, the development-associated Wnt/β-catenin pathway, the hypoxia-associated HIF1/2-mediated response and stress-associated pathways, among others [56,58]. Prototypical tumor suppressors and oncoproteins, such as TP53, PTEN, RB, BRCA1, MYC and MDM2 are SUMO targets [54].

Regarding the oncogenic protein MYC, contradicting results indicating that its PIAS1-mediated sumoylation leads to either transcriptional repression and proteasomal degradation, as to enhanced transcriptional activity, have been published (discussed in [56,57]). Besides PIAS1, SENP7 has been involved in the control of MYC sumoylation [59], and, more recently, SENP1 has also been indicated to de-sumoylate and stabilize MYC [60]. However, in relation to cancer, a relevant observation on MYC is that MYC-driven tumorigenesis requires sumoylation, since downregulation of SUMO E1 is lethal for tumor cells overexpressing MYC [61,62]. The tumor suppressor TP53 is a key regulator of the cell cycle, apoptosis and senescence, involved in DDR. Stability of TP53 largely depends on MDM2, which ubiquitylates TP53, targeting it for proteasomal degradation. In this context, sumoylation-mediated stabilization of MDM2 has been reported to favor TP53 degradation [63]. However, in turn, TP53 activity is also regulated by SUMO. A large variety of proteins have been observed to regulate TP53 activity in a SUMO-dependent manner, including several SUMO ligases and other proteins directly involved in modification of TP53 or of TP53 associated factors. They include TOPORS, RanBP2, CBX4, some TRIM members, MDM2, certain viral proteins and all PIAS members [31], although reports on the effect of these proteins on TP53 activity are in some cases controversial, for instance regarding PIAS4 (discussed in [30,31]).

Genome stability depends on accurate DNA replication and repair and on successful chromosome segregation, which is linked to proper cell cycle progression. PCNA is a SUMO target essential for DNA replication and repair, but sumoylation also regulates a variety of proteins involved in DNA damage sensing and repair [56]. For instance, sumoylated bloom syndrome helicase (BLM) can interact and promote the activity of RAD51 at damaged replication forks. Localization of the deubiquitylating enzyme Ataxin-3 to DNA double strand breaks (DSBs) is SUMO-dependent, and key DDR proteins like BRCA1 and BARD1 also operate in a SUMO-dependent manner [56,64,65]. In this scenario, the SUMO-targeted ubiquitin ligase RNF4 and the SUMO ligases PIAS1 and PIAS4 play essential roles. Besides, SENP7 has been implicated in chromatin relaxation for homologous recombination-dependent DNA repair [66], and SUMO ligase activity of the SLX4 complex has been differentially involved in the response to global and local replication stress [67].

Two main effects are observed in cancer cells when blocking sumoylation: the reduction of their proliferative capacity and the induction of the antitumor immune response [57]. Sumoylation participates in the dynamic regulation of a variety of targets in virtually all the phases of the cell cycle. Early studies on UBC9 depletion evidenced the fundamental role that sumoylation plays in cell cycle progression [17]. Defective sumoylation leads to aneuploidy and chromatin bridge formation. The spindle assembly checkpoint (SAC) guarantees the delay of anaphase to complete alignment of chromosomes on the mitotic spindle. Incorrect alignment halts mitosis through inhibition of the anaphase promoting complex/cyclosome (APC/C)-CDC20 by the mitotic checkpoint complex (MCC), and these complexes contain subunits targeted by SUMO [57]. Topoisomerase IIα (TOPIIα) is involved in decatenation of chromosomes prior to segregation and its activity is controlled by SUMO, which is also involved in recruiting BLM and polo-like kinase 1-interacting checkpoint helicase (PICH) to ultrafine chromosome bridges for resolution. SENP7 has been indicated to play an important role in keeping mitosis timing [25], but the other SENPs have also been involved in regulating different aspects of the cell cycle [68]. Among them, SENP3 plays a prominent role through phosphorylation-mediated inactivation of its desumoylating activity [69]. SENP3 is a redox sensor subjected to ARF-mediated proteasomal degradation, which acts on proteins like TP53 and MDM2 and modulates HIF1α activity via P300 desumoylation, and associates with a variety of human diseases, including cancer [70].

SUMO is also critical for the immune response, which is linked to inflammation and cancer. Important roles of SUMO in regulating key immune response pathways as NF-κB and interferon pathways have been clearly established [71]. In the case of interferon, TRIM proteins play an important role in mediating sumoylation of interferon regulatory factors (IRFs) [41]. Besides, activity of the immune system-associated factor signal transducer and activator of transcription 5 (STAT5) is also under the control of sumoylation (reviewed in [72]), as well as the cGAS-STING pathway involved in sensing of cytosolic DNA [42,73], and the inflammasome [74,75]. In relation to immunity, a number of oncogenic viruses have demonstrated to hijack the sumoylation pathway for their own benefit, participating in infection, persistence and transformation of host cells [76]. Among them are hepatitis B and C viruses, human papillomavirus, Epstein–Barr virus, Kaposi’s sarcoma-associated herpesviruses, human T-cell leukemia virus type 1 and Merkel cell polyomavirus. Interestingly, a variety of viral proteins may display SUMO ligase activity on viral and host targets [31].

We have already explained some examples involving SUMO-mediated regulation of specific transcription factors, dysregulation of which is linked to cancer. On top of that, SUMO has general chromatin-associated roles related to the safeguarding of transcriptional programs, alteration of which may also lead to cancer. This connects SUMO to epigenetic regulation. Indeed, among PTMs participating in the histone code is sumoylation [77]. Therefore, SUMO has been indicated to govern transcription by preserving appropriate chromatin states associated with specific transcriptional programs. For instance, it has been shown that SUMO preserves somatic and pluripotent cell identities through the enforcement of different chromatin states [78], and, furthermore, SUMO modification pattern of chromatin-associated repressive factors differentiates somatic from pluripotent cells [79]. Thus, impaired sumoylation, depending on the cell type, facilitates reprogramming to pluripotency, conversion of pluripotent to totipotent cells, transdifferentiation or direct differentiation [78]. All this suggests the involvement of SUMO in limiting cell-fate transitions. According to this, it has been also shown that altered sumoylation impairs neuronal differentiation [80], while SENP7 depletion in mouse embryonic fibroblasts (MEFs) facilitates reprogramming to induced pluripotent stem cells (iPSCs) [81]. Interestingly, it has been indicated that sumoylation probably plays a role in protecting the stability and functionality of transcription programs and signaling pathways susceptible of easy misregulation in tumorigenesis [54].

As a general observation, sumoylation pathway components appear upregulated in most cancers, correlating with metastasis, higher histological grade and stage, and poor prognosis. Few exceptions include SENP2 and PIAS2, which consistently show downregulated expression in cancer cells and correlate with better prognosis when overexpressed [58]. A number of factors influence expression and activity of pathway components. These comprise altered DNA methylation of component coding locus, altered expression of component-targeting miRNAs, altered transcription factor activity on component-coding genes, mutations in coding or regulatory regions associated with components and environmental factors, like hypoxia, presence of ROS, stress situations and toxic compounds [58]. As indicated, SENPs have been shown to be particularly sensitive to ROS and hypoxia conditions. Despite putative beneficial effects of acting globally against sumoylation for challenging cancer cells, non-desirable effects are also expected, since inhibiting global sumoylation does not distinguish between cancer and normal cells. Moreover, in general, sumoylation cannot be considered oncogenic or tumor suppressive. This will depend on the affected factor, pathway and/or cell/tissue. Thus, to fight cancer through sumoylation, it is crucial to selectively act on the modification process, and indeed, proteases and ligases are the pathway components mainly involved in target selection. Both groups of proteins have been related to different aspects of cancer [31,68], and the accurate dissection of their action mechanism is necessary for the establishment of selective therapeutic inhibition.

## 4. SUMO Proteases in Cancer

All SENPs have been associated with cancer (Table 2). Moreover, the more recently discovered protease USPL1 could also be linked to cancer, as it has been revealed to be critical for cell proliferation [23]. Among SENPs, SENP1 dysregulation is associated with a great variety of cancers [72]. Since it is the family pioneering member, it is also probably the most studied SENP protein.

### 4.1. SENP1

SENP1 is important for tumor development and drug resistance. Its overexpression has been reported in many cancers, including prostate (PCa) [82], colorectal (CRC) [83], breast [60,84], lung [85] and thyroid [86] cancers. In fact, its overexpression in these cancers has been a target point for some studies as a prognostic marker. However, SENP1 seems not to be overexpressed for instance in pancreatic cancer [87]. SENP1 is known to desumoylate and activate several proteins involved in promoting growth, migration, and evasion of cancer cells, like c-JUN, PIN1 and GLI1, among others [84,88,89]. It has been shown that it also enhances proliferation and migration in triple-negative breast cancer (TNBC) cells [90], as well as migration and invasion in neuroblastoma cells [91], through regulating expression of E-cadherin (CDH1), matrix metalloproteinase 2 (MMP2) and matrix metalloproteinase 9 (MMP9) proteins [91].

SENP1 is a key TP53 desumoylating enzyme and a prospective therapeutic target in cancer cells having wild-type TP53. SUMOylated TP53 could get easier access to target DNA and to coactivators contributing to TP53-mediated transcription [92].

In hepatocellular carcinoma (HCC), UBE2T desumoylation by SENP1 is related to carcinogenesis, which associates with upregulation of both proteins. Patients with increased SENP1 expression show larger and more tumors, with poorer histological characteristics and later stage in the TNM staging system, correlating with a lower overall survival rate [93].

The Warburg effect describes how most cancer cells prefer aerobic glycolysis over oxidative phosphorylation. To meet the nutrient requirements for rapid growth and proliferation, they need large quantities of nutrients, particularly glucose. Human hexokinase 2 (HK2) is a glycolysis regulator that links metabolic and proliferative functions in various cancers such as breast and lung cancers [94], HCC [95] and PCa [96]. HK2 can be sumoylated at K315 and K492, which are desumoylated by SENP1. A sumoylation mutant of HK2 preferentially attaches to mitochondria, increasing glucose consumption and lactate generation, while decreasing mitochondrial respiration. Of note, this metabolic reprogramming promotes PCa cell proliferation [97].

**Table 2 ijms-23-08012-t002:** SUMO proteases involved in cancer.

Proteases	Cancer Types *	References
SENP1	breast	[60,84,90,94]
	CRC	[83,98,99,100,101,102]
	HCC	[92,93]
	lung	[85,94]
	MM	[103]
	neuroblastoma	[91]
	ovarian	[104]
	pancreas	[87]
	PCa	[82,97,105]
	thyroid	[86]
SENP2	bladder	[106]
	breast	[107]
	GC	[108]
	HCC	[109]
	MM	[72]
	PCa	[110]
SENP3	AML	[111]
	breast	[112,113]
	CRC	[114,115]
	GC	[116]
	HCC	[117,118]
	HNC	[119]
	laryngeal	[120]
	OSCC	[121]
	osteosarcoma	[122]
	ovarian	[123]
SENP5	breast	[124]
	HCC	[125,126]
	OSCC	[127,128]
	osteosarcoma	[129]
SENP6	AML	[130]
	B-cell lymphoma	[131]
SENP7	breast	[26,132]
	CRC	[133,134]
	HCC	[135]
	PCa	[136]

* See abbreviations list.

A positive correlation has been observed between the expression of *SENP1* and *HIF1A* genes in patients with CRC, in which high levels of expression of both markers seem to be related to poor prognosis and possibly to drug resistance [98]. SENP1 has proven to be a novel target of miR-193a-5p, which has been reported to be sponged by the long noncoding RNA MCM3AP-AS1, and the MCM3AP-AS1/miR-193a-5p/SENP1 regulatory axis has been suggested as a potential therapeutic target in CRC treatment [83]. Furthermore, it has been published that miR-198-mediated suppression of SENP1 impacts cell proliferation, sphere and tube formation, and apoptosis in CRC cells [99]. Moreover, SENP1 suppression produced by miR-133a-3p has been also observed to limit ability of CRC cells to proliferate [102]. On the other hand, SENP1 has been shown to desumoylate MYC in cancer tissues, which can be suppressed by Momordin Ic (MI), resulting in cell cycle arrest and apoptosis in CRC cells [137]. Therefore, there are several publications suggesting SENP1 as a contributor to colorectal carcinogenesis [83,98,101], so that SENP1 could be a good candidate for the detection and treatment of human CRC. Moreover, MI increases SUMOylated protein levels and inhibits cell proliferation in PCa cells by interacting with SENP1. Therefore, MI as an inhibitor of SENP1 also has potential value to treat PCa [105].

It has been reported that there is an association between SENP1 expression and cancer immunity, suggesting that SENP1 could be used as a cancer immunotherapy target. SENP1 also correlates with immune infiltration, and a variety of transcription factors involved in regulation of apoptosis, proliferation, cell cycle, tumorigenesis, invasion and metastasis are responsible of the increased expression of SENP1 in cancer cells [138]. Among them is YY1, which, a few years ago, was shown to form a complex with the chromatin adaptor BRD4 to regulate *Senp1* expression [139]. Recently, it has been published that the E3 ubiquitin ligase SMURF2 is lowly expressed in CRC, and when overexpressed it targets YY1 for degradation, which results in inhibition of the SENP1/MYC axis and thereby in cancer prevention [100]. SENP1 depletion in multiple myeloma (MM) cells has been shown to decrease viability and proliferation, while inducing apoptosis, which correlates with inactivation of NF-κB signaling [103]. SENP1 also shows high correlation with sensitivity and resistance to anti-cancer drug and drug targeted genes [138]. For instance, SENP1 has been shown to reduce cisplatin sensitivity of hypoxic ovarian cancer cells [104].

To sum up, evaluation of SENP1 for specific cancer diagnosis, and, in particular, for CRC, and for making treatment decisions, is highly recommended. In addition to MI, other natural compounds, such as triptolide, together with a number of synthetic molecules, have demonstrated to be effective for SENP1 inhibition in a variety of cancer models [58].

### 4.2. SENP3 and SENP7

We have recently shown that, in cancer cells, SENP7 is downregulated under oxygen and glucose deprivation (OGD) conditions [133], which is a feature of inner cells inside solid tumors. SENP3 protein has also been reported to be degraded under OGD through the unfolded protein response kinase PERK and the cathepsin B enzyme [140]. Under OGD, overexpression of SENP7 is promoting cell survival, while downregulation leads to apoptosis. Interestingly, and in agreement with a previous report [140], SENP3 led to the opposite effect. This also correlated with the observation of prevalent amplification and deletion of the *SENP7* and *SENP3* genomic loci, respectively, in most cancer types. In particular in CRC, high *SENP7* expression associates with poor prognosis and higher transformation degree. High *SENP7* expression also associates with advanced tumor stage, which inversely correlates with *SENP3* expression. Therefore, cancer cells may escape from harmful OGD conditions by upregulating *SENP7*, which emerges as a putative prognosis marker in CRC [133]. Unlike SENP7, which seems to favor tumorigenesis, SENP3 has been reported to mediate antitumor functions [115].

As mentioned, SENP7 possesses two tandem motifs for HP1 interaction [25], which assure HP1 enrichment at pericentric heterochromatin [141]. The loss of one of these motifs due to alternative splicing generates the truncated isoform SENP7S, associated with better prognosis in breast cancer in comparison with the full-length isoform SENP7L [26]. In fact, in human mammary epithelia, *SENP7S* is the most abundant transcript, although it is considerably reduced in precancerous ductal carcinoma and in all breast cancer subtypes. SENP7S exhibits unaltered SUMO isopeptidase activity, but unlike SENP7L, it is found in the cytosol. According to this, SENP7S is ineffective in desumoylating HP1α when overexpressed. However, both sumoylated β-catenin and AXIN1 are SENP7S substrates for the maintenance of normal mammary epithelium physiology [132]. Of note, the ubiquitin ligase cullin 3 adaptor SPOP, which appears frequently mutated in a variety of cancers, has been shown to promote senescence by targeting SENP7 for degradation, which results in HP1α increased sumoylation and, thereby, in epigenetic gene silencing [136]. Furthermore, when comparing prostate tumor specimens with SPOP mutations to those with wild-type SPOP, SENP7 is expressed at higher levels, and SENP7 depletion causes PCa cells to become senescent [136]. Thus, inhibition of SENP7 could be used as a treatment for tumors with SPOP mutations. SPOP overexpression also promotes downregulation of vimentin associated with SENP7 degradation, which ultimately suppresses HCC cell migration and invasion [135]. A reduced number of synthetic compounds have shown to be effective against SENP7 and SENP3, but most of them exhibit cross-reactivity against other SENPs [58,68].

Proteasomal degradation also participates in SENP3 regulation. Under normal conditions, SENP3 is continuously degraded by the ubiquitin-proteasome pathway, and oxidative stress blocks degradation [142]. Oxidative stress is mainly achieved by production of ROS. Although elevated ROS levels are associated with different physiological and pathological situations, such as growth factor stimulation, inflammation and ischemia/reperfusion, cumulative evidence has established that ROS levels are particularly elevated in cancer cells due to oncogene activation, lack of blood supply or additional factors. Of note, ROS are tightly linked to EMT induction [143]. A moderate increase in ROS levels leads to SENP3 stabilization and relocalization from the nucleolus to the nucleoplasm, where SENP3 desumoylating activity is required for ROS-induced HIF1α transactivation [142]. As mentioned, this occurs through desumoylation of the HIF1α coactivator P300, which enhances partner binding. Furthermore, nuclear factor erythroid 2-related factor 2 (NRF2) is found in the nuclei of laryngeal carcinoma cells but not in cells from the surrounding tissues, which was found to be connected with the appearance of SENP3 in the nucleus. NRF2 promotes cell protection against oxidative stress, chemotherapy and radiotherapy, and it has been observed that cisplatin-induced ROS generation triggers intranuclear activation of NRF2 via SENP3 activity, which may be related to diminished cancer cell responses to chemotherapy [120]. Redistribution of SENP3 from the nucleolus to the nucleoplasm has also been reported in oral squamous cell carcinoma (OSCC) [121], as well as in OSCC, and SENP5 has been shown to be stabilized by ROS (see below) [127]. Compared to SENP1 and SENP2, SENP3 appears to be more responsive to mild oxidative stress [142]. It has been suggested that a moderate increase in intracellular ROS production can cause SENP3 stabilization by oxidation of C243 or C274, which would, therefore, block ubiquitin-mediated degradation. However, when an excessive amount of ROS are produced, SENP3 become inactive, due to oxidation of C532, which is required for substrate binding [144]. In breast cancer, high levels of SENP3 are linked to high levels of E2F targets, high tumor grade and poor prognosis. SENP3 interacts with E2F1, and oxidative stress inhibits interaction, resulting in enhanced sumoylation of E2F1, which modulates its transcriptional activity to enhance cell cycle arrest. This allows cells to resolve acute oxidative damage before SENP3-mediated desumoylation and reactivation of E2F1, which leads to cell cycle resumption [112]. SENP3 overexpression in cancer cells causes improper desumoylation of specific proteins, which impairs their function. Among them is PML, which is desumoylated by stabilized SENP3 in response to mild oxidative stress, diminishing its ability to limit cell growth [114]. SENP3 is accumulated in a number of primary human malignancies, which includes ovarian, lung and CRC cancers, and in this latter, PML appears hypo-sumoylated [114]. In relation to ovarian cancer, SENP3 expression was found to be strongly connected with stage, grade and lymph node metastases, as well as with a poor prognosis [123]. STAT3 hyperphosphorylation has been discovered in a variety of human malignancies, including head and neck cancer (HNC). Exposure of HNC cells to tobacco extract results in a quick Y705 phosphorylation of STAT3, but also in a rapid rise of SENP3, both of which were dependent on an increase in ROS [119]. SENP3 can enhance STAT3 phosphorylation through removing SUMO2/3 from K451, which impairs SIM-mediated association with the phosphatase TC45. In human laryngeal patients, a correlation between SENP3 protein levels and STAT3 phosphorylation has been observed, which is more significant in smokers’ specimens [119].

Despite evidence of positive correlation between SENP3 levels and malignancy in a variety of cancers, it has been conversely reported that SENP3 loss also associates with tumor progression in other cancer types. Oxidative stress-mediated driving of SENP3 activity is necessary for STING-induced antitumor action in dendritic cells (DCs). SENP3 depletion in DCs promotes tumor growth by suppressing antitumor immune response by dampening STING-dependent type-I interferon (IFN) signaling. ROS-mediated stabilization of SENP3 enables interaction with IFI204 and IFI16, which are desumoylated and boost STING signaling, which was assessed in CRC tissue [115]. Similarly, SENP3 loss associates with tumor progression in breast cancer. Tumor associated macrophages are critical for tumor progression. They mostly polarize to the M2 subtype, which displays anti-inflammatory and pro-tumor features. SENP3 depletion leads to increased AKT1 sumoylation in the breast cancer microenvironment, promoting macrophage polarization to the M2 subtype, which stimulates tumor growth and lymphatic metastasis. Furthermore, in breast cancer biopsies, SENP3 deficiency shows a strong negative correlation with expression of M2 markers in a variety of breast cancer subtypes, including TNBC, suggesting that SENP3 can be effective in suppressing M2 polarization [113]. Similar to SENP3, it has also been recently reported that SENP7 senses oxidative stress to sustain CD8^+^ T cell metabolism and antitumor function [134]. Thus, SENP7 depletion in CRC cells results in attenuated proliferation and impaired antitumoral function. T cell receptor signaling-mediated production of ROS in CD8^+^ T cells trigger cytosolic translocation of SENP7, which mediates PTEN desumoylation and degradation, preventing PTEN-associated metabolic defects [134]. On the other hand, SENP7 has been described as a regulator of inflammation [74,75], and inflammation associates with cancer. Indeed, a number of SENP7 interactors, identified under inflammatory conditions, are cancer related proteins [145].

Correlation between SENP3 expression and gastric cancer (GC) metastasis has also been reported [116]. Mechanistically, overexpressed SENP3 in GC cells targets the EMT-inducing transcription factor forkhead box C2 (FOXC2) for desumoylation, which, in turn, activates expression of the EMT-associated and transendothelial migration-promoting N-cadherin gene (*CDH2*), among others. In relation to this, it has also been shown that SENP3 overexpression in osteosarcoma cells results in inhibited expression of the epithelium-associated gene *CDH1*, which courses with promoter hypermethylation [122]. Database analysis indicates that in general *SENP3* is abundantly expressed in sarcoma and high expression correlates with poor prognosis, being shown in osteosarcoma that SENP3 depletion results in impaired proliferation, migration and invasion, and in enhanced apoptosis. Thus, SENP3 has been suggested as a powerful biomarker for osteosarcoma diagnosis [122]. *SENP3*, in combination with *ARID1A* and *CSMD* has also been proposed as an effective prognosis marker in HCC, despite molecular heterogeneity detected in HCC patients [118]. In HCC, it has also been shown that SENP3 overexpression significantly reduces isoflurane-mediated stimulation of sumoylation, resulting in decreased proliferation and invasion of HCC cells [117].

Two important processes, the dysregulation of which is tightly associated with cancer, are under the control of SENP3: mitosis and DDR. Strickingly, another PTM, phosphorylation, has been shown to be critical for regulation of SENP3 desumoylating activity during mitosis [69]. SENP3 phosphorylation during mitosis impairs its SUMO catalytic activity on proteins associated to chromosomes, which includes TOPIIα. CDK1 and PP1α were identified as the respective kinase and phosphatase involved in inactivation at mitosis onset and reactivation at mitosis exit, a process that when impeded leads to mitotic arrest and chromosome instability in U2OS cells. Moreover, U2OS cells expressing non-phosphorylatable SENP3 promoted tumor growth in nude mice. Interestingly, all nine mitotic phosphorylation sites on SENP3 are found in the noncatalytic N-terminal region of the protein [69]. Expression of the DSBs-repairing protein human coilin-interacting nuclear ATPase (hCINAP) correlates with acute myeloid leukemia (AML) prognosis. This protein is recruited to damage sites at late stage for resolution of DDR by promoting SENP3-mediated desumoylation of Nucleophosmin (NPM1), which is required in the sumoylated state at early stage for recruitment of DNA repair proteins [111]. Importantly, depletion of hCINAP sensitized to chemotherapy an AML patient-derived xenograft mouse model.

### 4.3. Other SENP Proteins

The MAPK kinase 5-extracellular-signal-regulated kinase 5 (MEK5-ERK5) pathway is critical for regulation of proliferation and survival in cancer cells. ERK5 is sumoylated in response to induced phosphorylation of MEK5 and overexpression of the HSP90 co-chaperone CDC37, participating in nuclear translocation of ERK5, which promotes proliferation of PCa PC-3 cells [110]. In this context, SENP2 overexpression abrogates epidermal growth factor-mediated nuclear localization of ERK5, suggesting that targeting ERK5 sumoylation can be of interest for fighting PCa [110]. SENP2 expression was shown to be significantly reduced in bortezomib-resistant MM patient samples, while other human SENPs are not [72]. As a result, it has been indicated that the decrease of SENP2 expression could have a deleterious effect on bortezomib-induced cell cycle inhibition and apoptosis, resulting in development of resistance. Thus, SENP2 overexpression sensitizes MM cells to bortezomib, while depletion of SENP2 results in opposite effect, correlating with increased sumoylation of IκBa and thereby in activation of NF-κB [72]. Similarly, in breast cancer, it has also been shown that SENP2 inactivates NF-κB, sensitizing cancer cells to doxorubicin [107]. It is commonly acknowledged that stemness is important for cancer metastasis and recurrence, contributing as well to drug resistance, resulting in poor patient prognosis. Overexpression of SENP2 has been shown to diminish HCC stemness and to sensitize cells to sorafenib [109]. HCC cell lines have lower SENP2 levels than normal human liver epithelial cell lines, while HCC stem cells had much lower levels than regular HCC cells. Mechanistically, SENP2 overexpression results in inhibition of the AKT/GSK3β/CTNNB1 pathway, which is linked to stemness suppression and drug sensitivity increase of HCC cells [109]. Meanwhile, a study with bladder cancer cells revealed that overexpression of SENP2 suppresses the TGFβ pathway and the ensuing EMT, preventing cell invasion. This effect relies, at least partially, on SENP2-mediated desumoylation of TGFβ receptor I (TGFβRI) [106]. Additionally, N-Myc downstream-regulated gene 2 (NDRG2) protein is a tumor suppressor and a SUMO target, which is frequently downregulated in several cancer types. NDRG2 degradation occurs through sumoylation-mediated ubiquitination targeting by the action of RNF4. In GC cells, it has been shown that overexpression of SENP2 desumoylates and stabilizes NDRG2, which results in inhibition of cancer cell proliferation [108]. Ebselen and some other synthetic compounds have shown efficacy against SENP2 [58]. In addition, SENP2 is also sensitive to additional non-specific SENP inhibitors [68].

SENP5 has been reported to play critical roles in progression of several cancers including breast cancer, osteosarcoma and OSCC. The first substrate of SENP5 identified was the tumor suppressor PML, which has an essential role in the regulation of cell proliferation [146], and SENP5 function has been associated with cytokinesis and the safeguard of mitochondria functionality; the latter being related to cytoplasmic desumoylation of DRP1 protein by SENP5, which inhibits mitochondria fragmentation [147,148]. SENP5 is overexpressed in osteosarcoma cells and its depletion causes cell growth inhibition and enhanced apoptosis [129]. In comparison with paracarcinoma epithelial cells, SENP5 is better expressed in OSCC, where it mostly localizes to the cytoplasm [128]. It is, in particular, expressed in the cells located at the inner layer of carcinoma nests and its expression has been linked to OSCC differentiation but not to any other clinicopathological factors. Similar to SENP3, mild oxidative stress stabilizes SENP5 in CAL-27 cells, but does not enhance apoptosis, whereas combined SENP5 depletion and mild oxidative stress led to mitochondria fragmentation and significantly increased cell apoptosis [127]. Database analysis indicates that breast cancer patients with low *SENP5* expression have a better prognosis. Indeed, SENP5 depletion inhibits proliferation, anchorage-independent growth, migration and invasion of breast cancer cells [124]. Absence of SENP5 results in destabilization of hypersumoylated TGFβRI, which correlates with reduced expression of the invasion-associated gene *MMP9*, and, thereby, with impeded metastasis. In HCC samples, *SENP5* was shown to be overexpressed as well, and its silencing results in decreased HCC cells proliferation [125]. SENP5-mediated tumorigenesis in HCC has been explained on the basis of the control that SENP5 exerts on ATR activation during DDR. In this context, it has been indicated that SENP5-mediated desumoylation of the ATR activator ATRIP leads to improper ATR activation during DDR [125]. Saikosaponin-d has been shown to reduce the malignance, in particular, Sonic Hedgehod-mediated EMT, of HCC cells, while boosting their sensitivity to other drug systems under hypoxia, through specific activation of SENP5, which is associated with GLI1 desumoylation [126].

It has been shown that *SENP6* is among the top 20 mutated genes in AML, appearing also significantly overexpressed compared to normal control samples, and showing sensitivity to I-BET-762 and Tubastatin A [130]. On the other hand, it has been described that *SENP6* is frequently deleted in lymphomas, which leads to unrestricted sumoylation [131]. Thus, in this scenario, SENP6 has been classified as a tumor suppressor. In MYC-driven B-cell lymphoma, depletion of SENP6 leads to dissociation from chromatin of protein complex involved in DNA repair and genome maintenance, resulting in damage accumulation and genome instability [131]. Few and non-specific compounds target SENP6 and SENP5 to some extent [58].

## 5. SUMO Ligases in Cancer

Although protein sumoylation can take place without the presence of ligases, these enzymes bring specificity and efficiency to the process. As mentioned, the most studied SUMO ligases are those of the PIAS family, whose involvement in cancer has been well studied [30,31]. Dysregulation of these enzymes is altered in many cancers, being detectable at both the mRNA and protein levels. Their interaction with various tumor suppressors as well as oncogenes promotes tumorigenesis and cancer cell survival. Besides PIASs, additional ligases, such as RanBP2 and CBX4, are key to cancer, the latter being involved in a variety of cancer types [56] (Table 3).

PIAS proteins, in addition to mediate sumoylation, are able to regulate the function of other proteins that act as transcription factors by inhibiting their binding to DNA, recruiting HDACs and through sequestration in nuclear foci or in the nuclear periphery. It has been described the interaction of these proteins with up to 60 cell members, affecting various cellular processes including DNA repair, immune regulation, cell proliferation and survival [30]. TP53 is able to interact with all members of the PIAS family [149]. The consequences of its sumoylation are diverse, showing some studies that this PTM contributes to its activation and others report that it leads to its nuclear export, thus counteracting its transcriptional activity [16,149,150]. Moreover, it has also been suggested that TP53 PIAS-mediated activation or repression may occur independently of its E3 ligase activity [149]. Like TP53, other members of the same family are also susceptible to PIAS-mediated sumoylation such as TP63, TP73 and the TP53 ubiquitin ligase MDM2 [149], thereby modulating the activity of all of them. PIAS-mediated sumoylation is also important in PI3K/AKT signaling, which plays an important role in the regulation of various physiological and oncogenic processes [151]. Sumoylation of AKT by PIAS1 causes its activation and deletion of the target K decreases the tumorigenic capacity of the E17K cancer-associated mutation [152]. In contrast, PIAS2α inhibits PI3K/AKT signaling through sumoylation of the tumor suppressor PTEN, thereby decreasing its degradation and inhibiting PI3K/AKT signaling through its phosphatase activity [153].

**Table 3 ijms-23-08012-t003:** SUMO ligases involved in cancer.

Families	Proteins	Cancer Types *	References
**SP-RING**	PIAS1	APL	[154]
		breast	[155,156,157,158]
		CRC	[159]
		EC	[160]
		GC	[161]
		GBM	[162]
		lung	[154,163]
		lymphoma	[61,164]
		melanoma, renal	[165]
		MM	[103,166]
		oncoviruses	[167,168,169]
		OSSC	[170]
		PCa	[171,172,173]
		thyroid	[174]
	PIAS2	breast	[175,176]
		GC, HCC, leukemia, ovarian, renal, sarcoma, testicular	[165]
		osteosarcoma	[177]
		PCa	[171]
		thyroid	[174]
	PIAS3	adrenocortical, mesothelioma, renal, sarcoma	[165]
		breast	[176,178,179,180,181,182]
		cervical	[183]
		CRC	[184,185,186]
		esophageal, glioma, lung	[187]
		GC	[169]
		GBM	[162,188,189]
		HCC	[190]
		OSCC	[162,188,189]
		PCa	[171,172,191,192,193]
	PIAS4	adrenocortical, mesothelioma, thyoma	[165]
		breast	[176,194,195]
		CRC	[196]
		GC	[197]
		HCC	[198]
		lung	[199,200]
		ovarian	[201]
		pancreas	[202]
**Others**	CBX4	bladder, cholangiocarcinoma, EC, esophageal, melanoma, mesothelioma, pancreatic, renal, sarcoma, thyoma, thyroid	[165]
		breast	[203]
		cervical	[204,205]
		CRC	[206]
		GC	[207]
		HCC	[208]
		lung	[209]
		osteosarcoma	[210]
		PCa	[211,212]
	RanBP2	cervical	[213]
		cholangiocarcinoma	[214]
		GBM	[162]
		HCC	[215]
		lung	[216]
		OSCC	[170]
		PCa	[217]
	BCA2	breast	[47]
	ZNF451	breast	[218]
		HCC	[219]
		pancreas	[220]
		PCa	[221]
	TRIM family	APL	[222]
		breast	[170,223,224,225]
		cervical	[226]
		CRC	[227,228,229]
		EC	[230]
		esophageal squamous cell	[231]
		GC	[232,233,234,235,236,237]
		HCC	[238]
		lung	[227,239,240,241,242,243]
		ovarian	[244,245,246]
		pancreas	[247]
		PCa	[248]
		renal	[249]
		thyroid	[250]

* See abbreviations list.

Not all proteins promoting sumoylation of specific targets, and, thereby, described as putative ligases, have been properly demonstrated to display such an enzymatic activity at the biochemical level. Furthermore, all of them have additional functions independent of SUMO. In the next sections, we focus on a few ligases, those that have been studied more in depth and have been demonstrated to associate with cancer in a SUMO-dependent manner.

### 5.1. PIAS1

As indicated, one of the first proteins described as sumoylatable was the tumor suppressor PML [251]. In APL, chromosomal translocation leading to the PML-RARα fusion oncoprotein [55] decreases PML NBs integrity [252]. Mono- or oligo-sumoylation via PIAS1 of this oncoprotein at K160 [154], causes its stabilization, essential for leukemogenesis [253]. However, treatment with arsenic trioxide (ATO), causes conformational changes in the protein that lead to its poly-sumoyation, enabling the binding to the E3 ubiquitin ligase RNF4 and, thus, its proteasomal degradation [254]. This results in APL cell death and ultimately in disease remission [255]. PIAS1, therefore, plays a role in both the development and remediation of APL. It has been observed that PML sumoylation by PIAS1 also causes the recruitment of casein kinase 2 (CK2), which phosphorylates PML leading to its ubiquitination and degradation [154]. *PIAS1* is overexpressed in non-small-cell lung cancer (NSCLC) cells and other cancers [154,184]. The expression levels of PIAS1 and PML inversely correlate in NSCLC samples, showing that PIAS1 depletion increases PML expression and decreases proliferation [154]. In addition, PIAS1 also mediates the sumoylation of Focal Adhesion Kinase (FAK), a regulator of cytoskeletal remodeling, mitogenic signaling and cell survival, inducing its autophosphorylation and activation [256], as well as regulating its subcellular localization [163]. PIAS1-mediated nuclear recruitment of FAK promotes DNA damage repair, which may be advantageous for the survival of genomically unstable NSCLC tumors. FAK is not only overexpressed in lung cancer, but also in other tumor types, such as breast, pancreatic and CRC [257], and is a positive regulator of EMT [163,257]. Therefore, its activation by PIAS1 would induce EMT. However, other studies show that PIAS1 negatively regulates EMT by antagonizing the activity of TGFβ, another EMT activator [258], so further studies are needed to resolve this controversy.

PIAS1 is involved in MYC-driven B-cell lymphomas. MYC induces UBC9 and PIAS1 expression, which promotes hypersumoylation in P493-6 B lymphoma cells and in several Burkitt’s lymphoma cell lines. Furthermore, PIAS1-induced MYC sumoylation stabilizes MYC and facilitates MYC phosphorylation-associated transactivation in B-cell lymphomas. Depletion of PIAS1 induces apoptosis in several MYC-driven lymphoma cell lines and prevents B-cell lymphomagenesis in mouse xenografts models [61,164]. Overexpression of PIAS1, along with other components of the SUMO pathway, has also been observed in MM patients as well as in MM cell lines [103,166]. MYC is also overexpressed in one third of breast cancers [259] and PIAS1 is essential for the viability of MYC-dependent breast cancer cells, with reduced proliferation of MYC-dependent MDA-MB-231 cell line observed when PIAS1 is depleted, which is not observed in MYC-independent MCF7 cells [60]. However, although MYC sumoylation appears to stabilize MYC, other studies indicate that such PTM causes recruitment of RNF4, leading to its ubiquitination and degradation [59]. Therefore, contradictory results have been published in relation to MYC and its sumoylation. Moreover, in relation to breast cancer, it has been observed that in hormone receptor (HR) positive breast cancer, the sumoylation pathway is an essential regulator of estrogen receptor (ER) α [260]. In the presence of hormone, both PIAS1 and PIAS3 are able to sumoylate ERα, as well as its cofactors, regulating ERα transactivation, which can also occur via PIAS1 and PIAS3 in a sumoylation-independent manner. Whereas the overexpression of PIAS1 has been shown to result in epigenetic silencing of breast cancer-associated genes, including *ESR1* (ERα) [158], PIAS3 stimulates the proliferation of ER-positive breast cancer cells, and acts as a co-activator to regulate NR2E3-mediated activation of *ESR1* expression [181,182], thus playing opposite roles in ERα regulation. The PIAS1 epigenetic pathway is increased in breast cancer and, in addition to silencing genes such as *ESR1*, also silences the tumor suppressor *WNT5A*, which has been shown to increase self-renewal of breast tumor initiating cells and requires PIAS1 activity as a SUMO E3 ligase [158]. PIAS1 is also involved in the sumoylation of the transcriptional regulator SKIL (SNON), which inhibits TGFβ-induced EMT in ER-positive MCF7 cell-derived organoids [156], as well as in TNBC MDA-MB-231 cell-derived organoids [155,156,157]. In addition, it is also able to inhibit activation of MMP2 in TGFβ-treated MDA-MB-231 cells, in which PIAS1 depletion causes morphological changes associated with invasion [157]. While PIAS3 has pro-proliferative effects in ER-positive breast cancer cells, in the MDA-MB-231 cell line it is able to inhibit the proliferation and EMT like PIAS1 [178,182], sumoylating in that case the E3 ubiquitin ligase SMURF2, and facilitating TGFβ receptor degradation [178]. PIAS1 also mediates the sumoylation of RUNX family transcriptional factors. It has been shown that PIAS1-mediated sumoylation of RUNX3, promoted by AKT kinase, inhibits its transactivation activity. It is also noteworthy that some RUNX1 mutants associated with breast cancer cannot be sumoylated through PIAS1, and that in sumoylation defective mutants of RUNX3 the tumor suppressor capacity is nullified, promoting tumor growth. This suggests that PIAS1-mediated sumoylation of RUNX plays a role in regulating the tumor suppressor activity of these proteins [261].

PIAS1, along with PIAS2 and PIAS3, are implicated in PCa promotion through inhibition of the CDK inhibitor P21 and modulation of androgen receptor (AR) signaling [171,191]. PIAS family members act as co-regulators that selectively increase or repress transcription of AR target genes through both sumoylation-dependent or independent mechanisms [171,172,262]. While PIAS1- and PIAS2-mediated AR sumoylation represses its transcriptional activity [262], PIAS1, independently of its SUMO ligase activity, can increase AR-mediated upregulation of growth-promoting genes to drive PCa cell proliferation [172]. However, a recent study has shown that PIAS1-mediated AR sumoylation leads to its translocation to the cytosol and subsequent degradation, and that this also requires sumoylation of PIAS1 with SUMO3, which, in addition to being essential for translocation, is also required for AR ubiquitination and degradation through recruitment of the ubiquitin E3 ligase MDM2 [173]. For its part, PIAS3 sumoylates STAT5, a regulator of PCa cell growth and stability [263] inhibiting its phosphorylation associated activation in COS-1 cells [193]. Additionally, a recent study has shown that the overexpression of the receptor tyrosine kinase-like orphan receptor 2 (ROR2), decreased in PCa patients, would suppress miR-199a-5p levels, increasing PIAS3 expression and consequently downregulating AKT2 and AKT phosphorylation, which leads to the inhibition of tumor cell migration and invasion [192]. Therefore, these studies show ambiguous roles for PIAS1 in PCa progression, whereas PIAS3 seems to unequivocally inhibit tumor growth.

PIAS1 also plays a role in cancers produced by oncoviruses. The transcription factor Forkhead box M1 (FOXM1), involved in cell cycle progression, cell proliferation and response to DNA damage, is dysregulated in several malignancies [264]. FOXM1 sumoylation results in translocation to the cytoplasm, where it is degraded, inhibiting its transcriptional activity [265,266]. The viral oncoprotein E7, which belongs to the human papillomavirus responsible for causing cervical cancer as well as genital cancers such as vulvar, vaginal, anal and penile cancers [267], interferes with FOXM1 sumoylation, protecting FOXM1 from degradation [265], which increases cell proliferation [266]. The infection with Epstein–Barr virus (EBV) is also associated with the development of cancers like nasopharyngeal cancer, Hodgkin’s and non-Hodgkin’s lymphoma and a variety of GC types [268], and among the expressed proteins involved in lytic reactivation to promote cell proliferation, viral production and oncogenesis is Rta [269]. The PIAS1-mediated sumoylation of this protein increases its transcriptional activity and thus lytic reactivation [167]. Meanwhile, a recent study has shown that the sumoylation of the sterile alpha motif and HD domain 1 (SAMHD1), a protein that hydrolyses deoxyribonucleoside triphosphates (dNTPs) and restricts infection of EBV and other viruses, promotes its anti-EBV activity increasing the association of this protein to the viral genome [168]. It, therefore, appears that PIAS1 may have opposing roles in virus replication, indicating that further studies are needed to understand its function and to use it as a therapeutic target to stop infection and thus associated tumorigenesis.

In GC, *PIAS1* is downregulated and has been shown to play a role in metastatic progression [161]. In the human GC cell line SCG7901, a study carried out under inflammatory conditions with IL-6, a cytokine that is secreted by cancer cells, has shown that overexpression of PIAS1 decreases the migratory and invasive capacity produced by IL-6 treatment in these cells, preventing transcriptional activation of the PI3K/AKT signaling pathway. Therefore, EMT initiation is inhibited [161]. PIAS3 is also downregulated by the micro-RNA miR-BART5-5p in GC associated with EBV infection, producing the activation of STAT3 and programmed death-ligand 1 (PD-L1). This leads to inhibition of apoptosis and increased cell proliferation, invasion, migration and immune scape [169]. However, contrary to PIAS1 and PIAS3, PIAS4 favors the survival of cancer cells, promoting the overexpression and stability of lysine demethylase 5B (KDM5B) in hypoxia through its sumoylation, thus provoking the inhibition of *CDKN1A* (P21) transcription [197]. *PIAS1* expression is also downregulated in endometrial cancer (EC), both at mRNA and protein levels. This is because miR-182-5p and miR-96-5p, upregulated in EC, cause *PIAS1* inhibition and consequently STAT3 activation, which, in turn, increases miR-182-5p and miR-96-5p expression and promotes malignant progression of EC [160].

In general, PIAS1 is also involved in tumorigenesis through the regulation of the alternative lengthening of telomeres (ALT) pathway, causing its activation and maintenance of telomere length, essential for the development and maintenance of cancer. This process takes place through the recruitment and activation of PIAS1 by the telomere-associated protein SLX4IP, causing the sumoylation of RAP1 and its translocation from the nucleus to the cytoplasm. There, RAP1 binds and activates IκB kinase (IKK), activating the transcription factor NF-κB and inducing *JAG1* expression, which promotes NOTCH-mediated signaling and ALT pathway [270].

### 5.2. PIAS2

PIAS2 is involved in the sumoylation of zinc finger homeobox 3 (ZFHX3) [175], a transcription factor that is active in several pathological processes including atrial fibrillation and carcinogenesis, as well as in circadian regulation and development [271,272,273]. PIAS2, both isoforms, is the only ligase capable of sumoylating this protein, disrupting its ubiquitination and degradation and, thus, increasing its stability. This PTM has been shown to be essential for promoting cell proliferation and tumor growth in the breast cancer line MDA-MB-231 [175]. However, physical interaction of ZFHX3 with PIAS3 decreases its sumoylation, avoiding the co-localization of ZFHX3 and SUMO1 in the nucleus [274]. In osteosarcoma PIAS2α is downregulated and its overexpression is able to repress the cell cycle inhibiting cyclin D1 and D3 [177]. PIAS2 is also dysregulated in thyroid cancer [174], and in other cancer types like GC, HCC, leukemia, ovarian, renal, sarcoma and testicular, it has been shown genetic alterations of this ligase that can contribute to cancer progression [165].

### 5.3. PIAS3

As mentioned above, PIAS3 plays a role in breast cancer. In addition to the functions previously mentioned, PIAS3 is also involved in the regulation of the receptor tyrosine kinase ERBB4 [180], whose intracellular domain released by proteolytic cleavage acts as a transcriptional co-regulator implicated in the regulation of mammary epithelial cell differentiation and proliferation [275,276,277]. PIAS3-mediated sumoylation of this intracellular domain promotes its accumulation in the nucleus, facilitating ERBB4 autokinase activity [180], and nuclear ERBB4 immunoreactivity is associated with worse survival of ER-positive patients compared to cell surface expression [275]. Another recent study has shown that the exosome-derived micro-RNA miR-181a is able to inhibit PIAS3, resulting in activation of the JAK/STAT signaling pathway that promotes the development of early-stage myeloid-derived suppressor cells, thereby accelerating tumor growth and immune escape [179].

PIAS3 is also implicated in the regulation of stem-like properties of glioblastoma (GBM) cells [188,189]. In GBM tissues PIAS3 is downregulated, and the SMAD6-induced ubiquitination-mediated degradation of PIAS3 induces STAT3-mediated proliferation and stem-like cell initiation [188], while inhibition of PIAS3 by TRIM8 by a similar mechanism maintains STAT3-mediated stemness and self-renewal in GBM stem-like cells [189]. However, PIAS3 overexpression has been shown to result in sumoylation of vimentin as a major target in these cells, leading to the inhibition of their invasive and migratory capacity, which points to vimentin as a therapeutic target [278]. Other studies also show that *PIAS3* is overexpressed in GBM samples, while the expression of *PIAS1* is downregulated, showing both, as well as other SUMO-related genes, such as *RANBP2*, single nucleotide variant (SNV) mutations [162] that are also observed in OSCC [170].

In cervical cancer samples, as well as in cell lines associated with cervical cancer, *PIAS3* is downregulated, while the microRNA miR-199a-5p that plays an oncogenic role by promoting cell proliferation, EMT and metastasis, shows high levels of expression. *PIAS3* is a direct target of miR-199a-5p and overexpression of *PIAS3* is able to reverse the effects caused by miR-199a-5p [183]. *PIAS3* also appears to be regulated by another micro-RNA, in this case miR-181b in CRC cells [185]. miR-181b activates STAT3, promoting the Warburg effect in CRC cells and CRC xenograft growth in mice, and does that by inhibiting *PIAS3* expression, as the Warburg effect and tumor growth are reversed with *PIAS3* overexpression [185]. However, another study shows that *PIAS3* is overexpressed in CRC samples [184], showing increased expression in more advanced stages of the disease [186].

In HCC tissues and cell lines, a low expression of *PIAS3* has been observed in comparison with healthy tissue. Oxidative stress is involved in tumor development, and in HepG2 cells, it has been shown that exposure to H_2_O_2_ decreases PIAS3 expression while increasing STAT3 expression and that overexpression of PIAS3 recovers the anti-oxidative response, decreases cell migration and invasion capacity [190]. Contrary to this, *PIAS4* is overexpressed in HCC, and higher expression correlates with a worse prognosis. In Huh-7 and HepG-2 cells, PIAS4 regulates AMPKα and NEMO sumoylation, promoting HCC proliferation, migration and invasion [198].

### 5.4. PIAS4

PIAS4 is involved in the sumoylation of HDAC1, an essential epigenetic regulator belonging to a conserved family of deacetylases that may be implicated in cancer progression [279]. In non-tumorigenic cells, SUMO1 sumoylation of HDAC1 mediated by PIAS4 leads to its ubiquitination and degradation. However, in breast cancer cell lines, where *PIAS4* is overexpressed, it has been observed that PIAS4 preferentially binds SUMO2 to HDAC1, protecting it from ubiquitination and degradation, and promoting its expression and activity that is involved in cancer progression [194]. Furthermore, in breast cancer, as well as in AML, a loss of function of CCAAT/enhancer binding protein δ(C/EBPδ), a protein that plays an important role in cell G0 growth arrest, has been observed. In HC11 non-transformed mammary epithelial cells, PIAS4 primarily, but also PIAS3 and PIAS2β to a lesser extent, repress the transcriptional activity of C/EBPδ, increasing cell proliferation/migration, which increases tumorigenesis. This PIAS4-mediated repression is independent of its SUMO E3 ligase activity and occurs through the interaction of the SAP domain with the transactivation domain of C/EBPδ [176]. In addition, PIAS4 also mediates the regulation of AMPK, involved in the inhibition of several cellular processes that are important for tumor progression. Sumoylation through PIAS4 of the AMPKα1 catalytic subunit inhibits AMPK activity through the mTORC1 signaling pathway [195]. Therefore, PIAS4 could be considered a therapeutic target in these cases.

PIAS4 has also been implicated in lung cancer. On the one hand, it has been observed that hypoxia-induced PIAS4-mediated sumoylation of the migration regulator SLUG increases its repressor capacity, decreasing the expression of target genes such as *CDH1* [200], promoting migration, invasion and metastasis in lung cancer. However, another study shows that *PIAS4* is downregulated in NSCLC, and that its overexpression inhibits tumorigenesis by increasing GATA2 sumoylation, which provokes the inhibition of its transcriptional activity that its essential for the survival of RAS-driven NSCLC cells [199].

In ovarian cancer, it has been reported that induction of PIAS4 by hypoxia prevents the binding of the transcriptional activator SP1 to the *SIRT1* promoter, inducing EMT activation. Therefore, restoration of *SIRT1* expression through targeting the sumoylation pathway could be a strategy to combat metastasis in ovarian cancer [201]. Induction of PIAS4 by hypoxia also causes it to interact with and suppress the tumor suppressor VHL, facilitating its oligomerization, which inhibits its function as a tumor suppressor in HIF1α-dependent and independent manners, contributing to overall tumor progression [280]. In PCa cell lines, where *PIAS4* expression is increased, its inhibition increases the expression of VHL and inhibits the expression of HIF1α and its target genes, showing that VHL expression is dependent on the SUMO E3 ligase activity of PIAS4 [202].

### 5.5. Other Ligases

CBX4, like PIAS ligases, shows different functions in different types of cancers. The tumor suppressor WW domain-containing oxidoreductase (WWOX) is downregulated in many tumors, such as breast, ovarian, GC, HCC and lung cancers [281,282,283,284,285]. The sumoylation of WWOX mediated by CBX4 promotes its suppressive activity on the oncogene *JUN*, and in the PCa cell line DU145, it has been demonstrated that decreases cell proliferation, indicating that this PTM is essential for its tumor suppressor activity [211]. However, negative functions have also been observed in relation to this SUMO E3 ligase, showing a recent study that CBX4 promotes cell growth and metastasis in PCa [212]. In HCC, CBX4 increases the transcriptional activity of HIF1α through its sumoylation at K391 and K477. This results in hypoxia-induced increased expression of vascular endothelial growth factor (*VEGF*) gene, and consequently in angiogenesis, regulating tumor cell proliferation, invasion and migration [208]. Nevertheless, another study has shown that sumoylation of HIF1α mediated by PIAS4, negatively regulates its stability and transactivation and downregulates VEGF-mediated angiogenesis, with a negative correlation between *PIAS4* expression and angiogenesis in CRC samples [196]. Additionally, in GC and lung cancer samples, where CBX4 is overexpressed, it has been observed its interaction with the oncogene *BMI1*, implicated in promoting cell growth, metastasis and stem-cell self-renewal. Accordingly, CBX4 depletion suppresses cell growth, migration and metastasis, suggesting that BMI1 may be involved in the CBX4 action mechanisms [207,209]. In breast cancer, CBX4-mediated sumoylation of hTERT, the catalytic component of the human telomerase enzyme, causes the retention of the hTERT/ZEB1 complex at the *CDH1* promoter, leading to its repression and then to EMT, promoting cancer cell migration and invasion [203]. In cervical cancer, as well as in CRC, CBX4 also shows a negative role, promoting cancer progression [204,205,206].

Regarding the SUMO ligase RanBP2, it has been observed its implication in the stability of insulin-like growth factor-1 receptor (IGF-1R) [286], involved in tumor growth and survival [287]. Nuclear IGF-1R is overexpressed in cancer cells [288] and high levels inversely correlate with survival of cancer patients [289,290]. The nuclear accumulation of IGF-1R requires interaction with RanBP2, which in turn requires IGF-1R sumoylation by RanBP2 [286]. RanBP2 is also involved in the cellular localization of the tumor suppressor P27KIP1. Thus, in the cholangiocarcinoma cell line QBC939, sumoylation of P27KIP1 contributes to cytosolic translocation, which impairs G1 cell cycle arrest and promotes cancer cell growth [214]. In HCC, RanBP2 interacts with CEBPα facilitating its sumoylation and degradation. This causes dysregulation of O-GlcNAcylation, controlled by O-GlcNAc transferase and O-GlcNAcase, being the latter downregulated, which results in hyper-O-GlcNAcylation of oncogenic proteins, such as proliferative-activated receptor gamma coactivator 1 alpha (PGC1α), promoting HCC progression [215]. In cervical cancer, it has also been observed that RanBP2 plays an oncogenic role [213].

Another SUMO E3 ligase is breast cancer associated gene 2 (BCA2), whose expression is increased in more than 50% of invasive breast cancers [291]. Studies on BCA2 show conflicting results, assigning to it both oncogenic [292] and tumor suppressor roles [293]. A recent study in breast cancer cells has shown that although BCA2 promotes the transition from G1 to S phase of the cell cycle, it does not increase cell proliferation, migration or metabolic activity of the cells. Furthermore, BCA2 overexpression decreases the activity of NF-κB, hyperactivated in cancer cells and regulating many genes involved in proliferation [47]. That inhibition occurs through BCA2-mediated sumoylation and stabilization of IκBa, which inhibits NF-κB [294]. Moreover, BCA2 regulates the tumor suppressor IRF1 independently of its SUMO E3 ligase activity, causing IRF1 activation in ER-positive and IRF1 downregulation in ER-negative breast cancers, which may explain the contradictory observations reported in relation to the role of BCA2 in breast cancer [47].

ZNF451 is also implicated in cancer and its expression levels are increased in breast cancer, HCC and pancreatic cancer [218,219,220]. It has been recently described that ZNF451 mediates TWIST2 sumoylation, which promotes its stability and consequently the activation of EMT, associated with cancer metastasis [219].

In relation to the TRIM family of proteins, many of them have SUMO E3 ligase activity as discussed above, and have been implicated in numerous cancer processes (reviewed in [31]). However, the relationship between their function as SUMO E3 ligase and the development of cancer is largely unknown. As an exception, TRIM33, similar to PIAS1, interacts with and sumoylates SKIL, suppressing TGFβ-induced EMT [43].

## 6. Conclusions

Virtually all cancer types are linked to dysregulation of components of the sumoylation pathway, which results in altered sumoylation. This is due to the involvement of SUMO regulation in many relevant processes related to tumorigenesis, such as cell cycle progression, genome replication and repair, transcription and metastasis. We have explained that in fighting cancer by acting on the SUMO pathway, in order to avoid undesirable general effects by globally inhibiting sumoylation, it is better to act specifically on SUMO proteases and ligases. These are the main actors involved in target selection, leading to local and defined alterations related to a precise type of cancer, thus making the difference between transformed and healthy cells. The latter would otherwise surely be severely affected by global inhibition. In most cancers, components of the sumoylation pathway appear upregulated; therefore, it makes sense that the best way to counteract this is through inhibitors able to titrate the excessive activity. However, despite the efforts made in identifying specific drugs against particular SUMO proteases and ligases, the most interesting drugs at present for SUMO inhibition are acting on E1, thus leading to global inhibition. A variety of natural compounds have been demonstrated to have an inhibitory effect on different components of the SUMO pathway, but many of them are not advantageous, since they frequently act in the micromolar range and not only target sumoylation. Conversely, synthetic drugs offer more selectivity and effectiveness. Among them, the most promising at present is TAK-981, which is structurally related to adenosine 5’-monophosphate and acts by forming a covalent and irreversible adduct with SUMO through the catalytic activity of E1 [57]. Successful results *in vitro* and in some animal models have led to the initiation of several phase I/II clinical trials with this drug for several types of cancer, and, in most cases, in combination with immunotherapy [58]. However, efforts in developing selective drugs continues. To date, no inhibitors have been identified against SUMO E3 ligases [57], opening up a field of study to enhance and increase the specificity of sumoylation inhibition, since different SUMO E3 ligases are responsible for sumoylation of different target proteins. As explained, ligases are quite different from each other in terms of structure and activity, and in many cases, they display a scaffold function rather than an enzymatic activity. In contrast to this, and similar to E1 and E2, proteases display a defined and well-characterized enzymatic activity easier to chemically counteract. Indeed, the greatest hopes for efficient inhibition of the sumoylation pathway in cancer are in general pinned on SENPs. Moreover, up to six different SENP proteins, exhibiting specific SUMO paralog preferences and target selection, together with singular cell type-dependent expression patterns and distinctive cancer-associated altered expression, make these proteins the best candidates to inhibit sumoylation in a targeted and selective manner. In this regard, SENP1 has sparked a great interest, since it has been solidly implicated in a number of cancer types. Nonetheless, there is still a long way to go, since the participation of many components of the pathway in tumorigenesis is still not well understood from a mechanistic point of view. In many cases, there is a clear correlation between altered expression of a pathway component and a particular cancer type, but ultimate SUMO targets, the dysregulation of which contributes to initiation or progression of tumorigenesis is largely unknown. Moreover, throughout the development of cancer, different pathway components may be involved, being beneficial to act against some of them, but detrimental to act against others. Thus, a detailed dissection of sumoylation action mechanisms is necessary to successfully exploit its inhibition in cancer therapies.

## Figures and Tables

**Figure 1 ijms-23-08012-f001:**
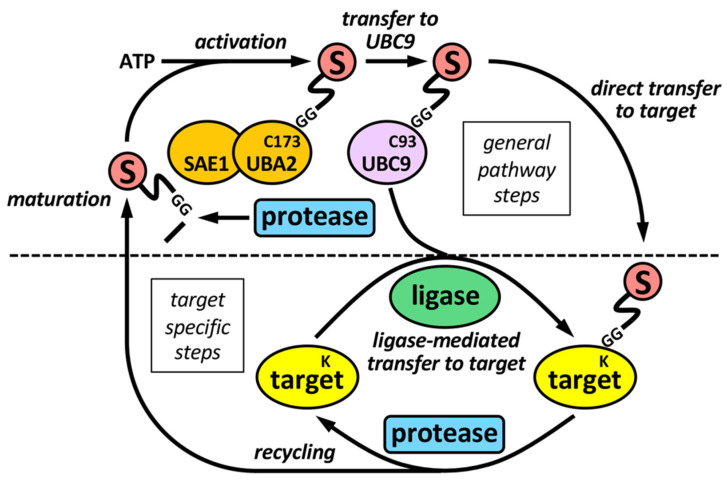
The sumoylation pathway. E1 enzyme (SAE1-UBA2 heterodimer), by using ATP, activates mature SUMO and transfers it to the E2 enzyme UBC9, which ultimately transfer SUMO to targets, either directly, or more frequently, assisted by an E3 SUMO ligase. SUMO maturation and scission from targets are performed by specific SUMO proteases. Maturation involves the exposition of two tandem Gly (G) residues at the C-terminus of mature SUMO. Activation and transfer to UBC9 involve the formation of thioester bonds between C-terminus of mature SUMO and specific Cys (C) residues at catalytic sites of E1 and E2, respectively. Defined Lys (K) residues at targets are the final SUMO acceptors for covalent attachment. Proteases and ligases are linked to target selection-associated steps in the sumoylation pathway.

**Figure 2 ijms-23-08012-f002:**
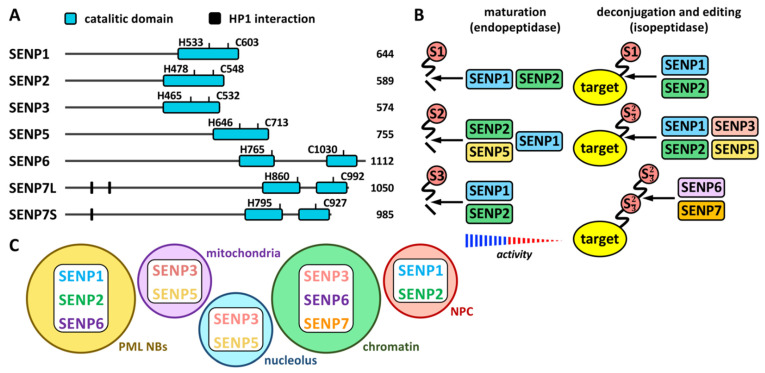
The SENP family. (**A**) Schematic representation of human SENP proteins with the C-terminal catalytic domain and the key His (H) and Cys (**C**) residues. HP1 interacting domains are also shown on SENP7, for which two isoforms have been described: large (SENP7L) and short (SENP7S), lacking the latter of one of the HP1 interacting motifs. The number of amino acids is also shown for each protein. (**B**) Endopeptidase (maturation) and isopeptidase (deconjugation and editing) activities for each SENP protein are shown. Strength of activity in relation to paralog preference is indicated for maturation activity. (**C**) Localization of SENP proteins to different cellular compartments is schematically represented. NPC, nuclear pore complex.

**Figure 3 ijms-23-08012-f003:**
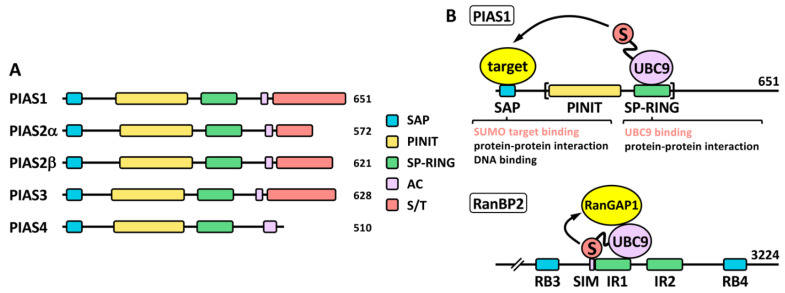
The PIAS family. (**A**) Schematic representation of human PIAS proteins. The SP-RING and SAP domains, together with the PINIT region, the acidic stretch (AC) and the C-terminal Ser/Thr (S/T) rich region, are shown. Two alternative isoforms, α and β, have been classically described for PIAS2. The number of amino acids is also shown for each protein. (**B**) Proposed ligase mechanisms using PIAS1 and RanBP2 as models. In the case of PIAS1, simultaneous binding of target and SUMO-loaded UBC9 through different domains may facilitate SUMO transfer to the target. The SP-RING is important for UBC9 binding but may also mediate other protein interactions for additional functions. Other domains, like the SAP domain, may recruit SUMO targets, but it is also involved in additional functions. The minimal protein region required for ligase activity is indicated with brackets. In the case of RanBP2, interaction of SUMO and UBC9 with a SIM and the 50-amino acid internal repeat (IR) 1, respectively, should position the thioester bond in the appropriate conformation for efficient transfer of SUMO to the target, which is not required to directly interact with the ligase. RB, Ran binding domain.

## Data Availability

Not applicable.

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
