# Peer review of "Cancer-Associated Dysregulation of Sumo Regulators: Proteases and Ligases"

_ijms, 2022, doi:10.3390/ijms23148012_

Round 1

Reviewer 1 Report

The authors compiled the information on SUMOylation, a type of post-translation modification in association with cancer progression specifically, keeping the respective proteases and ligases in consideration. There are many other reviews available on this topic; however, the authors have updated some information too. I have a few concerns and suggestions provided below.

Figure 1 is too generalised and can be substituted with an additional figure providing the molecular mechanism data. I understand that it is basic data, but please use the space in the manuscript while providing the new information. There are many good reviews already describing this information. Consider making other illustrations to report the detailed mechanism or could be a proposed mechanism. The main idea is to provide the knowledge which is not being compiled as the review form earlier.

Why the selection of only the PIAS family in this review? There are many more ligases involved.

Consider updating the SNEP data with more recent articles. Consider including this study as well: https://doi.org/10.1016/j.celrep.2019.11.028 and similar kind.

It could be better if authors could report the studies showing which ligase or protease could act as a drug target for cancer treatment.

Author Response

Reviewer 1

The authors compiled the information on SUMOylation, a type of post-translation modification in association with cancer progression specifically, keeping the respective proteases and ligases in consideration. There are many other reviews available on this topic; however, the authors have updated some information too. I have a few concerns and suggestions provided below.

Figure 1 is too generalised and can be substituted with an additional figure providing the molecular mechanism data. I understand that it is basic data, but please use the space in the manuscript while providing the new information. There are many good reviews already describing this information. Consider making other illustrations to report the detailed mechanism or could be a proposed mechanism. The main idea is to provide the knowledge which is not being compiled as the review form earlier.

Response: We agree with the Reviewer that Figure 1 is too basic and similar figures appear in many other reviews. We decided to display it because it helps to visualize the specific pathway steps where proteases and ligases act, and to separate general steps of the pathway from those that are target-specific, illustrating the idea that ligases and proteases are the main pathway components involved in target selection. We have now modified the figure to show this idea more clearly and to add additional information regarding the different steps of the process and the amino acid residues involved in catalytic activity. Additional mechanistic aspects of proteases and ligases are illustrated in Figures 2 and 3. In Figure 2 we have now indicated in the figure (not only in the legend) that maturation by proteases involves an endopeptidase activity, while SUMO recycling involves isopeptidase activity.

Why the selection of only the PIAS family in this review? There are many more ligases involved.

Response: We have explained in our review that to associate SUMO-related functions of ligases with cancer may be controversial, because most ligases have not been convincingly demonstrated to display a true enzymatic activity at the biochemical level, but also because all of them have additional functions not related with SUMO. This is the reason why we have focused on some ligases, not only on PIAS but also on CBX4, RanBP2, BCA2, ZNF451 and TRIM proteins, the latter ones grouped on a common section entitled “Other ligases”. This is due, as explained in the text, to the fact that PIAS proteins were the first discovered ligases, and consequently there is more information on them, not only linking them to cancer but specifically demonstrating the involvement of its SUMO-related function in cancer. So, to facilitate reading we decided to make subsections for the different PIAS proteins and to group the rest of relevant ligases in a final subsection. It is true that there are several dozens of proteins with putative ligase activity but formal demonstration of enzymatic activity at the biochemical level is needed. For most of them it has been described they display enhanced sumoylation effect on selected targets, which does not suffice to ascribe them a true enzymatic ligase activity. In addition, many of them, when altered, may contribute to cancer initiation or progression but whether this is associated with SUMO or with additional SUMO-independent roles has not been well addressed in most of the cases. For instance, dysregulation of many TRIM proteins is related to cancer, but few reports link this to the SUMO ligase activity of certain TRIM proteins. An exception is TRIM33, whose relation to cancer is also related to its SUMO ligase activity, as we explain in our manuscript. In any case, since the Reviewer has experienced problems with the way we present the data, we have modified the text of the introduction of this section and now explain better why we have focused in certain ligases (3rd paragraph of section 5).

Consider updating the SNEP data with more recent articles. Consider including this study as well: https://doi.org/10.1016/j.celrep.2019.11.028 and similar kind.

Response: Although this reference is not directly related to cancer it is a good example of the link between the SUMO pathway and inflammation, which in its turn is quite related to carcinogenesis. This reference also indicates that a number of proteins interacting with SENP7 under inflammation conditions are linked to cancer. Then, we have included this reference and others linking SENP7 with the inflammasome (end of 4th paragraph of subsection 4.2).

It could be better if authors could report the studies showing which ligase or protease could act as a drug target for cancer treatment.

Response: Unfortunately, no specific drugs have been developed to date against SUMO ligases, as stated in the manuscript. However, a number of natural and synthetic compounds have demonstrated to display inhibitory effects on the different SENP proteins, both in a specific and a general manner. Although we already mention some of them, we now make mention to additional compounds when describing each protease (subsections 4.1, 4.2 and 4.3).

Reviewer 2 Report

Overall, the review "Cancer-Associated Dysregulation of Sumo Regulators: Proteases and Ligases" by Lara-Ureña et al., is well organized and offers a comprehensive overview about SUMO proteases and ligases in cancer.

However, the manuscript would benefit from minor revisions, mostly regarding English language, adding references and correction of typos.

Some examples 

line 86-89 instead of "... SUMO ligases and proteases, involved in dynamizing changes in the sumoylation status of target proteins. In this review we update the current knowledge on how unbalance function of these pathway components impacts on cancer initiation and progression..." use "... SUMO ligases and proteases, which determine the sumoylation status of target proteins. In this review we summarize the current knowledge about deregulation of these enzymes in cancer and their impacts on tumor initiation and progression"

line 120-121 replace with " [125]. Also, a SENP7 splice variant, lacking one of these motifs, has been reported to predict for good prognosis in breast cancer patients."

line 123-132 add relevant references for this section

line 149 instead of "not related to", "independent of"

line 173 use "Initial in vivo studies using the yeast PIAS1 ortholog Siz1"

line 453 instead of "destroyed", "degraded"

line 932 remove "cancer cell"

line 958 "different from each other"

line 966 instead of "arouses" use "has sparked a"

And so on. These are just examples: check throughout the all manuscript.

All "in vitro" and "in vivo" should be in italics 

Make sure that all the abbreviations are specified either in the text or the end. Some are missing (i.e. TNM).

Tables: aligning text to the left and may improve readability

Author Response

Reviewer 2

Overall, the review "Cancer-Associated Dysregulation of Sumo Regulators: Proteases and Ligases" by Lara-Ureña et al., is well organized and offers a comprehensive overview about SUMO proteases and ligases in cancer.

However, the manuscript would benefit from minor revisions, mostly regarding English language, adding references and correction of typos.

Some examples 

- line 86-89 instead of "... SUMO ligases and proteases, involved in dynamizing changes in the sumoylation status of target proteins. In this review we update the current knowledge on how unbalance function of these pathway components impacts on cancer initiation and progression..." use "... SUMO ligases and proteases, which determine the sumoylation status of target proteins. In this review we summarize the current knowledge about deregulation of these enzymes in cancer and their impacts on tumor initiation and progression"

- line 120-121 replace with " [125]. Also, a SENP7 splice variant, lacking one of these motifs, has been reported to predict for good prognosis in breast cancer patients."

- line 123-132 add relevant references for this section

- line 149 instead of "not related to", "independent of"

- line 173 use "Initial in vivo studies using the yeast PIAS1 ortholog Siz1"

- line 453 instead of "destroyed", "degraded"

- line 932 remove "cancer cell"

- line 958 "different from each other"

- line 966 instead of "arouses" use "has sparked a"

And so on. These are just examples: check throughout the all manuscript.

Response: We have made all the corrections suggested by the Reviewer, and also, we have performed an additional complete checking for similar corrections in the rest of the manuscript. We also provide references for the indicated section as requested.

All "in vitro" and "in vivo" should be in italics 

Response: Al “in vitro” and “in vivo” have been changed to italics.

Make sure that all the abbreviations are specified either in the text or the end. Some are missing (i.e. TNM).

Response: TNM is not exactly an abbreviation but a system to evaluate staging of tumors, where T describes size, N spread to nearby regions and M metastasis. We have modified the corresponding sentence to avoid confusion. Regarding abbreviations, we initially decided not to include all of them in the final list to keep it to a minimum. However, we have revised all the abbreviations and included additional ones in the final list. We have not included in the list those terms not used a second time but abbreviated because they are widely known abbreviations (for instance ORF).

Tables: aligning text to the left and may improve readability

Response: We have aligned text in tables to the left to improve readability.

Round 2

Reviewer 1 Report

The authors have done good job answering the queries.